# Functional and metabolic fitness of human CD4⁺ T lymphocytes during metabolic stress

Lisa Holthaus[1,2,3] , Virag Sharma[3,5,6] , Daniel Brandt[1], Anette-Gabriele Ziegler[2,3,4], Martin Jastroch[1,7], Ezio Bonifacio[1,3,5,6]

**Human CD4⁺ T cells are essential mediators of immune responses. By altering the mitochondrial and metabolic states, we defined metabolic requirements of human CD4⁺ T cells for in vitro activation, expansion, and effector function. T-cell activation and proliferation were reduced by inhibiting oxidative phosphorylation, whereas early cytokine production was maintained by either OXPHOS or glycolytic activity. Glucose deprivation in the presence of mild mitochondrial stress markedly reduced all three T-cell functions, contrasting the exposure to resveratrol, an antioxidant and sirtuin-1 activator, which specifically inhibited cytokine production and T-cell proliferation, but not T-cell activation. Conditions that inhibited T-cell activation were associated with the down-regulation of 2′,5′-oligoadenylate synthetase genes via interferon response pathways. Our findings indicate that T-cell function is grossly impaired by stressors combined with nutrient deprivation, suggesting that correcting nutrient availability, metabolic stress, and/or the function of T cells in these conditions will improve the efficacy of T-cell–based therapies.**

## Introduction

T cells are quiescent surveillants that can rapidly transform into proliferative effector T cells to target pathogens or anomalous expression of self-proteins. This transition is tightly controlled at the signaling and metabolic levels. Understanding the control checkpoints and the requirements for T-cell–mediated responses are crucial in developing efficient, sustainable therapies that enhance or dampen T-cell–mediated pathologies or T-cell–based therapies.

The T-cell quiescent state is defined by low metabolic, transcriptional, and translational activity (Yang et al, 2011; Chapman et al, 2020). Quiescence is maintained by a cell-intrinsic program that includes IL-7 signaling (Kimura et al, 2013; Buck et al, 2015; Geltink et al, 2018) and extrinsic gatekeepers, such as tonic signaling (Kerdiles et al, 2009; Sprent & Surh, 2011; Chang et al, 2015; Chapman et al, 2020). Bioenergetically, resting T cells are defined by low metabolic demands, a reliance on oxidative phosphorylation (OXPHOS)–derived ATP, and suppression of glycolysis (O'Neill et al, 2016; Chapman et al, 2020).

T cells are activated when the TCR binds to a cognate major histocompatibility complex–peptide complex. This initiates a signaling cascade resulting in T-cell proliferation, differentiation, and effector function such as cytokine production (Smith-Garvin & Koretzky, 2009). TCR stimulation induces a glycolytic program that supports this response. Mitochondrial respiration is necessary for direct Ca²⁺ delivery to the immunological synapse during activation (Quintana et al, 2007; Menk et al, 2018). Mitochondrial reactive oxygen species (ROS) are crucial signaling molecules that induce nuclear factor of activated T cells, and subsequently IL-2 (Sena et al, 2013; Buck et al, 2015). The TCR binding strength determines whether T cells enter the cell cycle and proliferate, which is further modified by the level of mammalian target of rapamycin complex 1, and by activation and expression of IFN regulatory factor 4, and the proto-oncogene *MYC* (Wang et al, 2011; Man et al, 2013; Yang et al, 2013). The rapid clonal expansion that occurs after T-cell activation must be accompanied by changes in the nutrient and energy needs. Glucose and glutamine metabolism rapidly increase, favoring cell growth, clonal expansion, and survival (Dimeloe et al, 2017; Chapman et al, 2020). Costimulation via CD28 supports metabolic reprogramming, with a shift from OXPHOS towards an increase in glucose uptake via glucose transporter-1 and aerobic glycolysis. Mitochondrial respiration and other pathways, including one-carbon metabolism, the tricarboxylic acid (TCA) cycle, and lipid and sterol biosynthesis are augmented upon T-cell activation (Weinberg et al, 2015; Ron-Harel et al, 2016; Dimeloe et al, 2017; Tan et al, 2017; Chapman et al, 2020).

Although the normal physiological functions of T cells are well known, the performance of T cells, especially human T cells, under stressful conditions, such as nutrient deprivation or metabolic

[1]Institute for Diabetes and Obesity, Helmholtz Zentrum München, German Research Center for Environmental Health, Munich-Neuherberg, Germany    [2]Institute of Diabetes Research, Helmholtz Zentrum München, German Research Center for Environmental Health, Munich-Neuherberg, Germany    [3]German Center for Diabetes Research (DZD e.V.), Neuherberg, Germany    [4]Forschergruppe Diabetes e.V. at Helmholtz Zentrum München, German Research Center for Environmental Health, Munich-Neuherberg, Germany    [5]Center for Regenerative Therapies Dresden, Faculty of Medicine, Technische Universität Dresden, Dresden, Germany    [6]Paul Langerhans Institute Dresden of the Helmholtz Center Munich at the University Hospital and Faculty of Medicine of TU Dresden, Dresden, Germany    [7]Department of Molecular Biosciences, The Wenner-Gren Institute, Stockholm University, Stockholm, Sweden

Correspondence: ezio.bonifacio@tu-dresden.de

stress, is not fully understood. T cells, whether native or extrinsically administered often need to function despite compromised host conditions. Therefore, we investigated how metabolic and nutrient deprivation affects human CD4+ T-cell activation, cytokine production, and proliferation, and we identified pathways and genes affected by these conditions.

# Results

### Metabolic requirements of T-cell activation and proliferation

In murine cells, mitochondrial OXPHOS is generally associated with resting and activated T-cell states and glucose metabolism is associated with the effector T-cell functions. We confirmed and extended this general model of metabolic reprogramming by investigating the need for mitochondrial ATP, measured by OXPHOS activity. Human CD4+ T cells were first monitored after stimulation with anti-CD3/CD28 beads in the presence of 5 mM of glucose to define the activation phase as the first 16 h post-stimulation, and the transition phase to proliferation as 16–72 h post-stimulation (Fig S1A and B).

As expected, inhibition of mitochondrial ATP production by including oligomycin, an inhibitor of the $F_O$ unit of mitochondrial ATP synthase, during the activation phase (0–16 h) markedly reduced T-cell proliferation ($P$ = 0.03; Fig 1A). Unexpectedly, T-cell proliferation was also reduced if oligomycin was only present after the activation phase from 16 to 72 h ($P$ = 0.012; Fig 1B). These data suggest that OXPHOS is used by human T cells in both the activation

and the effector proliferation phases. Glucose uptake was increased 16 h after T-cell stimulation ($P$ = 0.012) and increased markedly after 72 h ($P$ = 0.033) (Fig 1C), consistent with an enhanced glucose requirement during T-cell proliferation. Mitochondrial ROS production was increased 10 min after stimulation ($P$ = 0.002), consistent with the suggestion that ROS are involved in the activation process (Sena et al, 2013), and was increased further at 16 and 72 h after activation (10 min versus 72 h: $P$ = 0.002; Fig 1D).

### Heterogeneous effects of mitochondrial and metabolic inhibitors on T-cell metabolism

We next examined oxygen consumption rate (OCR) and extracellular acidification rate (ECAR) during the proliferation phase. T cells were stimulated with anti-CD3/CD28 beads in the presence of 5 mM glucose with and without injection of metabolic inhibitors after 48 h of stimulation and OCR and ECAR measured using the xf extracellular flux analyzer (Figs 2 and S2B–G). In addition to oligomycin, we examined the effects of rotenone, an inhibitor of mitochondrial respiratory complex I. Resveratrol, was also included as it has multiple effects on cell metabolism such as inhibition of ROS and the $F_1$ subunit of the ATP synthase and activation of the protein deacetylase sirtuin 1 (SIRT1) (Howitz et al, 2003; Lagouge et al, 2006; Gledhill et al, 2007; Zou et al, 2013).

OCR in the bead-stimulated cells was reduced by the addition of oligomycin (1 μM, $P$ = 0.02; 2.5 μM $P$ = 0.031) or rotenone (1 μM, $P$ = 0.046; 2.5 μM, $P$ = 0.049) at 48 h, but was unaffected by resveratrol (Fig 2A–C). Many cell types compensate for impaired mitochondrial ATP production by increasing their glycolytic rate (Pavlova &

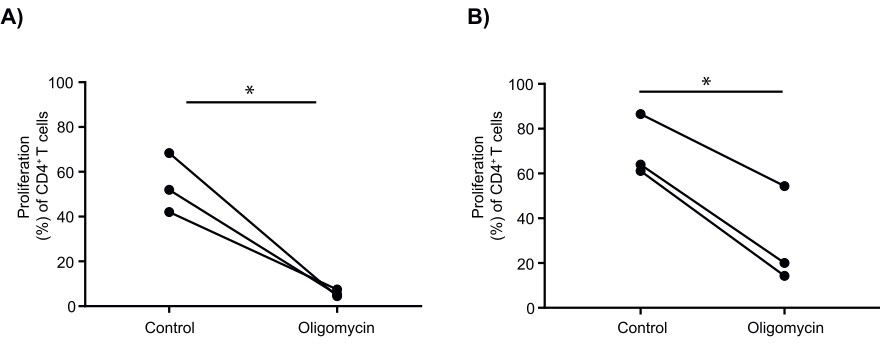

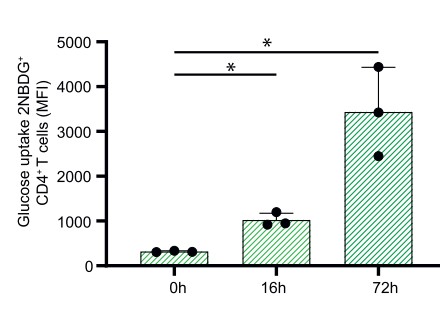

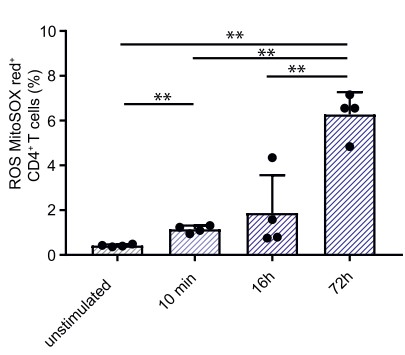

**Figure 1. Mitochondrial function and cellular glucose uptake in human CD4+ T cells.**
Human eFluor450-labeled CD4+ T cells were stimulated using anti-CD3/CD28 beads and cultured in the presence of 5 mM glucose. **(A, B)** Proliferation was measured by dye dilution at 72 h (A, B). **(B)** Oligomycin was either added to the cell culture together with the stimulation beads (time point 0 h; A), and then washed out after 16 h for the remainder of cell culture or (B) added after 16 h and kept in the medium until the end of culture. **(C)** Glucose uptake during CD4+ activation was monitored in anti-CD3/CD28 bead-stimulated T cells using 2-NBDG after 0, 16, and 72 h. Data are represented as mean fluorescence intensity and mean + SD; Each point represents one independent experiment. **(D)** Mitochondrial reactive oxygen species production was determined using MitoSOX Red in unstimulated T cells, and after 10 min, 16 h, and 72 h of stimulation. Comparisons were made using two-tailed paired $t$ tests in (A) and (B), repeated-measures one-way ANOVA in (C) and (D). *$P$ ≤ 0.05 and **$P$ ≤ 0.01.

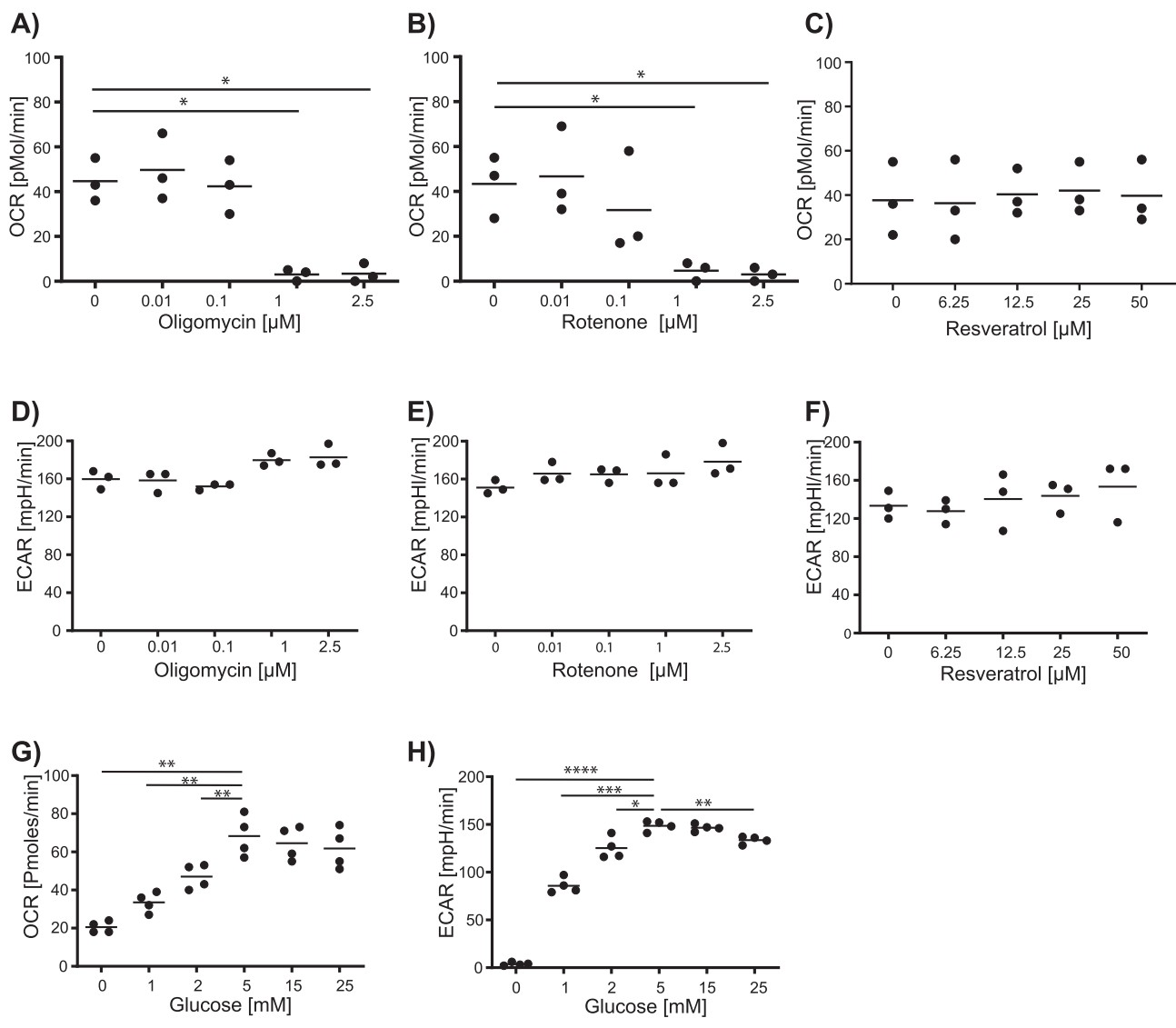

**Figure 2. Respiratory signatures of human CD4+ T cells in the presence of metabolic inhibitors.**
Human CD4+ T cells were stimulated with anti-CD3/CD28 beads for 48 h and the T-cell respiration signature was analyzed using an extracellular flux analyzer. **(A, B, C)** Mitochondrial ATP-linked respiration and **(D, E, F)** Glycolytic activity at the indicated concentrations of oligomycin, rotenone, and resveratrol added in real time during the measurement period in the presence of 5 mM glucose. **(G, H)** Effects of low and high glucose concentrations on mitochondrial ATP-linked respiration (panel G) and glycolytic activity (panel H). Comparisons were made using repeated-measured one-way ANOVA with Fisher's least significant difference test. *$P \leq 0.05$, **$P \leq 0.01$, ***$P \leq 0.001$, and ****$P \leq 0.0001$.

Thompson, 2016; Gaude et al, 2018). However, ECAR did not increase at oligomycin or rotenone concentrations that inhibited OCR (Fig 2D–F) suggesting maximal glycolytic rates already at uninhibited conditions. Glucose concentrations below 5 mM after stimulation for 48 h reduced both OCR (0 mM, $P = 0.001$; 1 mM, $P = 0.001$; 2 mM, $P = 0.005$ versus 5 mM glucose; Fig 2G) and ECAR (1 mM, $P = 0.0003$; 2 mM, $P = 0.013$), which was completely blocked in the absence of glucose ($P = 0.0001$) (Fig 2H).

To further determine effects of mitochondrial inhibition during the T-cell activation phase, glucose uptake and ROS production of bead-stimulated CD4+ T cells were examined after 16 h. Glucose uptake was reduced when cells were exposed to 1 μM oligomycin (mean fluorescence intensity [MFI], control: 1,021; oligomycin: 679.5, $P = 0.022$), 1 μM rotenone (MFI: 668.2, $P = 0.048$), or 50 μM resveratrol (734.5, $P = 0.053$) (Fig 3A). ROS production in unstimulated CD4+ T

cells was increased in the presence of 1 μM rotenone (MFI: 223 versus 61 in control conditions, $P = 0.0003$) or 1 μM oligomycin (MFI: 204, $P = 0.001$), but not in the presence of 50 μM resveratrol (MFI: 55, $P = 0.066$). ROS production was increased 16 h after bead stimulation (MFI: 176). ROS production was further increased in the presence of 1 μM rotenone (MFI: 290, $P = 0.041$) and 1 μM oligomycin (MFI: 272.2, $P = 0.003$). However, activation induced ROS production was blocked by 50 μM resveratrol (MFI: 84, $P = 0.006$ versus uninhibited) (Fig 3B). The glucose concentration did not affect glucose uptake or ROS production by T cells (Fig S3A–C).

In summary, during the activation phase, ROS production is increased by mitochondrial inhibition via oligomycin or rotenone, inhibited by resveratrol and unaffected by glucose deprivation, whereas glucose uptake is inhibited by oligomycin, rotenone,

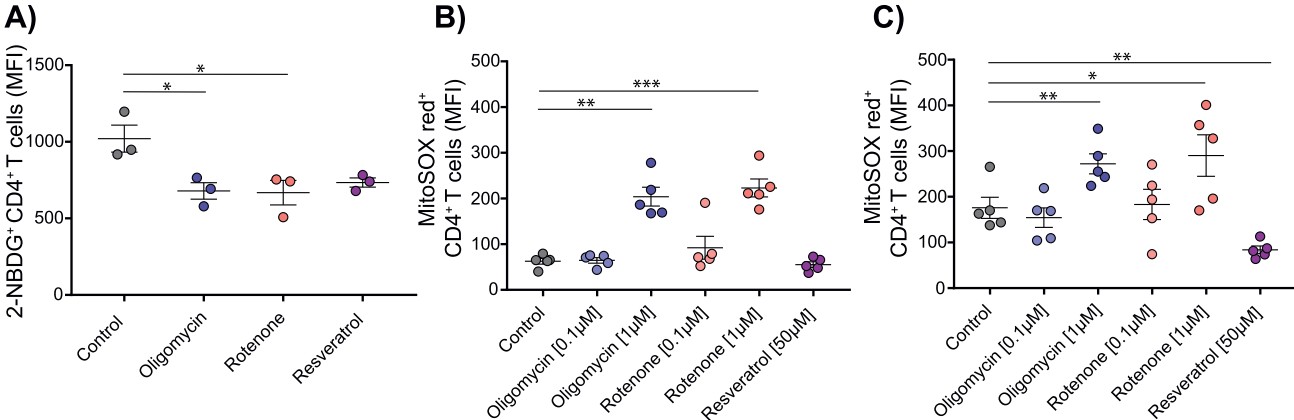

**Figure 3. Glucose uptake and reactive oxygen species production in the presence of metabolic inhibitors.**
**(A)** CD4⁺ T cells were stimulated with anti-CD3/CD28 beads for 16 h in the presence of 5 mM glucose with or without 1 μM oligomycin, 1 μM rotenone, or 50 μM resveratrol. The cells were starved for the last 2 h and labeled glucose (2-NBDG) was added to the culture for 10 min, after which glucose uptake was measured by flow cytometry. Data are shown as the mean ± SEM (n = 3) mean fluorescence intensity. **(B, C)** T cells were left unstimulated (panel B) or were stimulated with anti-CD3/CD28 beads for 16 h (panel C) in the absence (control) or presence of the indicated inhibitor. Mitochondrial reactive oxygen species production was monitored using MitoSOX Red by flow cytometry and is shown as the mean ± SEM MitoSOX Red mean fluorescence intensity (n = 5). Comparisons to the control conditions without inhibitor or 5 mM glucose were made using repeated-measures one-way ANOVA, with Fisher's least significant difference test *$P \leq 0.05$, **$P \leq 0.01$, and ***$P \leq 0.001$.

resveratrol, and glucose deprivation. During proliferation, oligomycin, rotenone, and glucose deprivation inhibit mitochondrial ATP-linked respiration without significant compensation via glycolysis.

### Effects of metabolic inhibitors on T-cell activation, proliferation, and cytokine production

Having established the effects of inhibition on parameters of cell metabolism, we next tried to link these metabolic alterations to T-cell function (Fig 4). CD4⁺ T-cell activation was measured as the frequency of cells positive for the early activation markers CD69 and CD25 16 h after bead stimulation in 5 mM glucose (Fig S4A). Exposure to oligomycin reduced CD69⁺ cells (mean: 54.1% versus control mean: 95.2%, $P = 0.045$) and CD25⁺ cells (mean: 56.7% versus control mean: 94%, $P = 0.042$) already at a concentration of 0.01 μM. T-cell activation was also reduced in cells exposed to rotenone at a concentration of 1 μM (CD69 mean: 57%, $P = 0.008$; CD25 mean: 49.4%, $P = 0.02$) (Fig 4A). Resveratrol or glucose deprivation had little or no effect on the frequency of activation marker-positive CD4⁺ T cells (Fig 4A and B). Taken together with the effects on metabolism, it is likely that mitochondrial ATP production, but not glucose uptake is essential for CD4⁺ T-cell activation.

The observation that T-cell activation was impaired by inhibiting ATP synthase at a lower concentration than was mitochondrial ATP production suggested an additional role for mitochondrial function in human T-cell activation beyond ATP production. Inhibition of the ATP synthase not only depletes mitochondrial ATP production but also alters redox balance by hyperpolarization. To investigate the importance of polarization changes for T-cell activation, we depolarized mitochondria in oligomycin-treated, activated CD4⁺ T cells by the addition of the uncoupler carbonyl cyanide-4-trifluoromethoxy phenylhydrazone (FCCP). Exposing oligomycin-treated T cells to FCCP partially rescued T-cell activation as measured by CD69⁺ on CD4⁺ T cells (Fig S5A and B), suggesting a role for mitochondrial polarization and ROS production in the activation process.

T-cell proliferation was analyzed at 72 h after bead stimulation (Figs 4C and S4B). Proliferation was inhibited by 0.01 μM oligomycin (mean dye-dim cell frequency: 35.5% versus 81.4% without inhibitor; $P = 0.004$) and by 1 μM rotenone (6.3%; $P = 0.002$). Proliferation was also reduced by resveratrol in a dose-dependent manner from a concentration of 12.5 μM (77.4% versus 93.9% without inhibitor; $P = 0.032$) and at low glucose concentrations (37.7% at 1 mM glucose, $P = 0.0001$). These data are also consistent with the requirement of glucose uptake for proliferation.

The production of cytokines (IFNγ, IL-21, IL-4, IL-17A, and IL-22) by CD4⁺ T cells was measured 6 h after bead stimulation (Nomura et al, 2000; Kaveh et al, 2012) (Fig S6). Oligomycin or culture in low or high glucose concentrations had minor and inconsistent effects on T-cell cytokine production (Figs 4C and S7A and C). Rotenone decreased IFNγ, IL-4, and IL-22 ($P = 0.018$) at 1 μM, but its effects were not dose-dependent (Fig S7B). These data indicate that cytokine production may use different energy sources because either restricting OXPHOS or glucose alone had little effect on this function. In contrast, resveratrol significantly and consistently decreased the production of IFNγ, IL-21, and IL-22 and these reductions were dose-dependent (Fig 4E).

### Low glucose and mitochondrial inhibitors synergistically affect T-cell function

Because cytokine production was unaffected by oligomycin or glucose restriction alone, we asked how restricting both OXPHOS and glucose impaired T-cell function. T-cell activation and T-cell proliferation were inhibited by the presence of oligomycin or rotenone with physiological or high glucose concentrations (5–25 mM) (Fig 5A and B). Glucose deprivation in the presence of either inhibitor markedly increased these effects of the inhibitors. Glucose deprivation reduced T-cell activation further in the presence of oligomycin from a mean value of 43.4% of CD69⁺ CD4⁺ T cells in 5 mM glucose to 0.9% in 0 mM glucose ($P = 0.046$; Fig 5A). Glucose

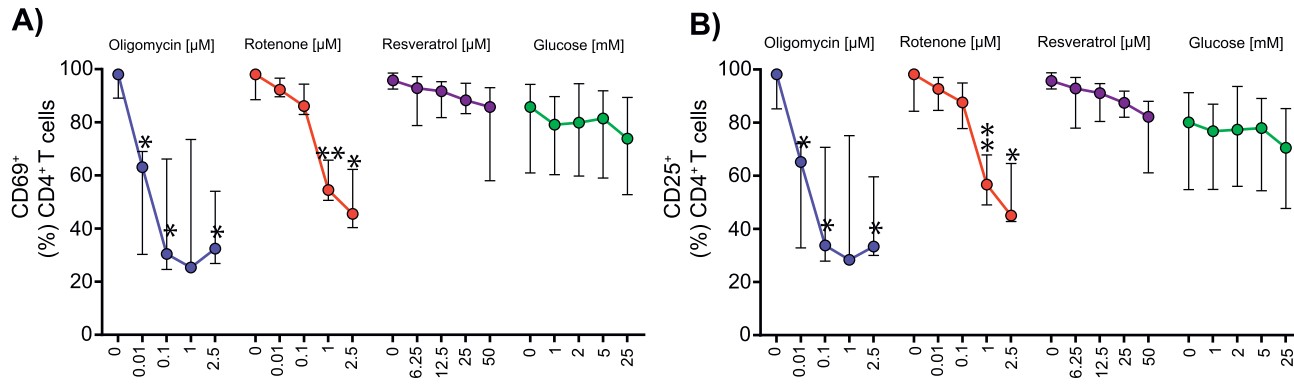

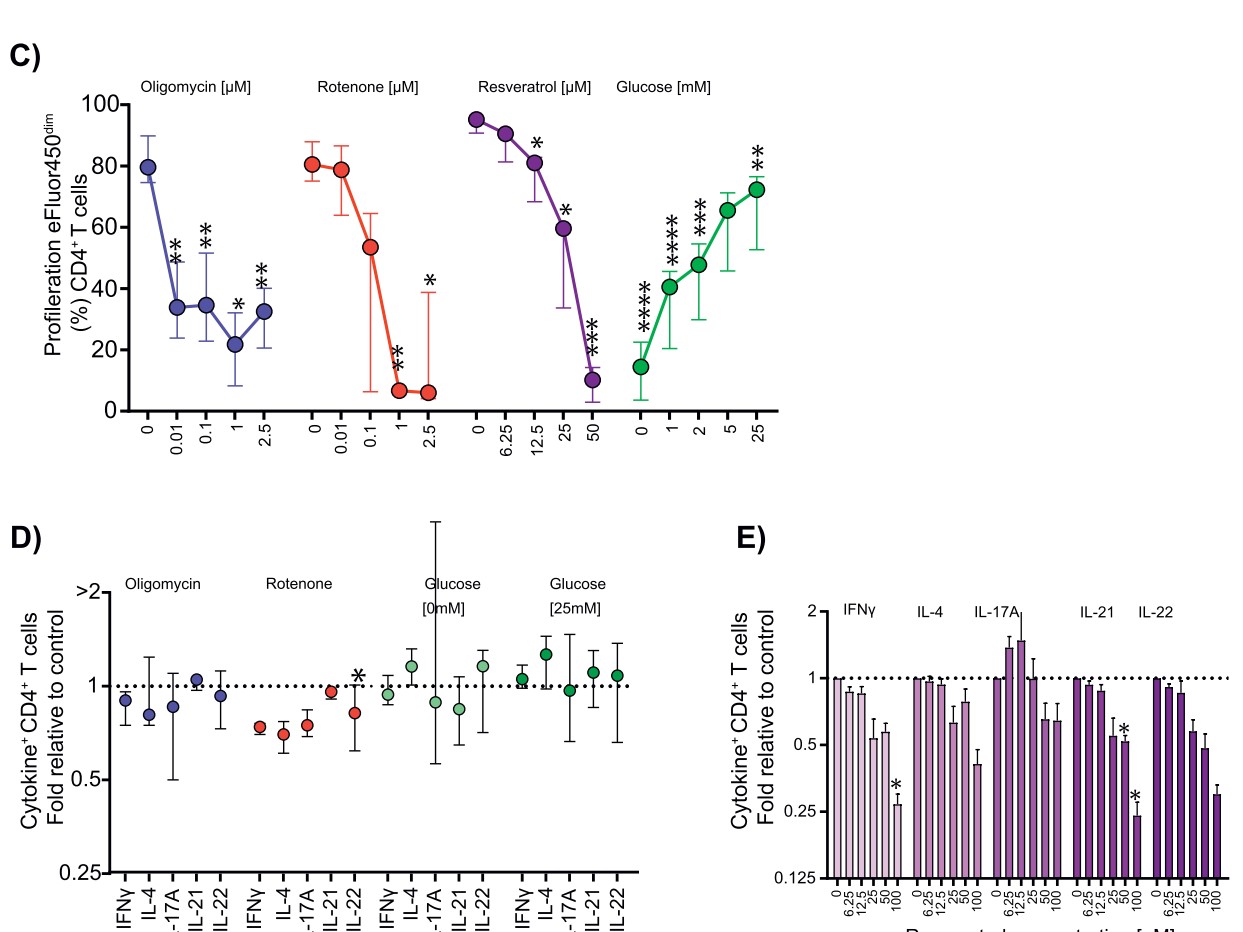

**Figure 4. Effects of metabolic perturbations on T-cell function.**
**(A, B)** T-cell activation markers CD69 (panel A) and CD25 (panel B) after 16 h of anti-CD3/CD28 stimulation in the absence (control) or at increasing doses of oligomycin (n = 3), rotenone (n = 3), and resveratrol (n = 4), each in the presence of 5 mM glucose, and in different glucose concentrations without additional inhibitor (n = 3). The surface marker expression of CD69 and CD25 is shown as the median with interquartile range of the frequency of positive cells T. **(C)** T-cell proliferation was measured by dye dilution at 72 h after stimulation of eFluor450-labeled CD4+ T cells in the absence or presence of oligomycin (n = 3), rotenone (n = 3), and resveratrol (n = 3) each in the presence of 5 mM glucose, and in different glucose concentrations (n = 8) without additional inhibitor. Data are shown as the median with interquartile range of the frequency of proliferated cells. **(D, E)** Intracellular cytokines were measured 6 h after stimulation with anti-CD3/CD28 beads in the absence or presence of oligomycin (1 $\mu$M) (n = 3), rotenone (1 $\mu$M) (n = 3), or resveratrol (50 $\mu$M) (n = 3), and in different glucose concentrations (n = 4) (panel D) as well as in different concentrations of resveratrol (panel E). Secretion inhibitor brefeldin A was added for the last 4 h. Data are shown as the relative fold value of the frequency of positive cells for each of the cytokines as compared to control conditions as median with interquartile range. Comparisons to the control conditions without inhibitor or 5 mM glucose were made using repeated-measures one-way ANOVA with individual variances computed for each comparison in (A, B, C), and paired $t$ tests in (D, E). *$P \leq 0.05$, **$P \leq 0.01$, ***$P \leq 0.001$, and ****$P \leq 0.0001$.

deprivation also caused a further reduction in T-cell proliferation when combined with rotenone (Fig 5B). Remarkably, although IFNγ production was unaffected by glucose deprivation alone (1 mM glucose, fold value = 1.01) or 0.1 μM oligomycin alone (fold value = 0.93), it was completely abrogated by glucose deprivation in the presence of 1 μM oligomycin (5 mM glucose and oligomycin versus 0 mM glucose and oligomycin: relative fold value, 0.02; $P$ = 0.02; Fig 5C). A reduction in IFNγ production was also observed following glucose deprivation in the presence of 1 μM rotenone (Fig 5C). These data, demonstrating synergistic action on each T-cell function by mitochondrial inhibition and glucose restriction, support the observation that both OXPHOS and glycolysis are relevant to proliferation and suggest that cytokine production is a relatively robust function that may be fueled by ATP derived from either OXPHOS or glycolysis.

## Identification of genes and pathways affected by short-term metabolic perturbation

Both metabolic inhibitors alone or in combination with glucose deprivation were associated with distinct impairments of T-cell functions. We used these effects to search for genes and pathways linked to specific T-cell functions during conditions that included the inhibition of T-cell activation (oligomycin or 1 rotenone), and cytokine production, but not T-cell activation (50 μM resveratrol). The transcriptome of CD45RO[+] (memory), and CD45RO[−] (naïve) CD4[+] T cells at 6 h after bead stimulation was sequenced and the genes that were differentially expressed in the presence versus absence of each inhibitor were identified (Fig S8A–C).

The pathways affected by conditions that most strongly inhibited the activation of memory CD4[+] T-cell inhibition (oligomycin) were the IFN signaling pathways, including the 2′,5′-oligoadenylate synthetase (OAS) pathways (Fig 6A). Multiple genes in these pathways were down-regulated by both oligomycin and rotenone, including *OAS1*, *OASL*, *OAS3*, and several IFN-induced genes, whereas *RNASEL* and the IFNγ-responsive genes *DPB1* and *TRIM2* were up-regulated (Fig 6B). Genes involved in the IFN pathway were also affected by oligomycin and rotenone in naive CD4[+] T cells. Oligomycin and rotenone strongly up-regulated several heat shock protein genes in naive and memory CD4[+] T cells (Fig S9).

Consistent with the effects on cytokine production, numerous cytokine genes were down-regulated by resveratrol in memory CD4[+] T cells and resveratrol affected the cytokine signaling pathway in memory CD4[+] T cells. Exposure to rotenone elicited weaker effects on this pathway. Interestingly, although the combination of oligomycin plus glucose deprivation markedly reduced cytokine secretion, there was relatively little effect on cytokine genes (Fig 6C), suggesting that resveratrol and oligomycin plus glucose deprivation differ in their path to cytokine secretion impairment. Resveratrol additionally affected genes involved in the interconversion of nucleotide diphosphates and triphosphates in memory CD4[+] T cells (Fig 6A).

## Resveratrol mimics the effect of sirtuin 1 (*SIRT1*) activation in human T cells

The effect of resveratrol on human T-cell function and metabolism suggested that resveratrol does not directly interfere with mitochondrial function, including the previously suggested inhibitory effect on ATP synthase or complex I of the mitochondrial respiratory chain. Another proposed resveratrol target is *SIRT1* (Howitz et al, 2003; Zou et al, 2013). RNAseq of 6 h bead-stimulated naive, memory, and regulatory T cells showed that the negative regulator of *SIRT1*, *HIC1*, was transcriptionally down-regulated by resveratrol in all three CD4[+] T-cell subsets and, despite only short-term exposure, also a concomitant increase in *SIRT1* and downstream genes such as *TP53* and *PPARA* in some of these T cells subsets (Fig 7A). We, therefore, examined effects on cells that are associated with *SIRT1* activation (Fig 7B–F). These effects include improved mitochondrial fitness (Lagouge et al, 2006; Price et al, 2012), caloric restriction (Guarente, 2013), and decreased mitochondrial ROS production (Hori et al, 2013). Consistent with these effects, we observed increased mitochondrial biogenesis in unstimulated CD4[+] T cells with increasing concentration of resveratrol (25 μM resveratrol; $P$ = 0.02) (Fig 7B and C); an increased spare respiratory capacity (25 μM resveratrol $P$ = 0.0013) (Fig 7D); decreased mitochondrial ROS production (25 μM resveratrol $P$ = 0.04) (Fig 7E), and a decrease in glucose uptake (72 h, $P$ = 0.047) (Fig 7F).

# Discussion

The interactions between immune signals and cell-intrinsic metabolic programs determine the functional status of a T lymphocyte. By altering the metabolic environment, we observed that all T-cell functions depend on cellular ATP levels, but each function differs in its preferential ATP source. The activation process mainly required mitochondrial-derived OXPHOS. The proliferation process receives ATP from glycolysis and OXPHOS. Cytokine production was able to switch ATP source according to the local environment. T-cell activation was impeded by the mitochondrial inhibitors oligomycin and rotenone but not by resveratrol, whereas cytokine production was decreased by resveratrol and by inhibiting both mitochondrial ATP and glucose availability. T-cell proliferation was reduced in all of the tested metabolic perturbations, emphasizing the high energetic demand of T-cell anabolism. Short-term treatment with oligomycin and rotenone was associated with down-regulation of genes in the IFN signaling pathways, including genes in the OAS pathway. Resveratrol altered the transcription of multiple genes in cytokine signaling pathways, as well as *HIC1*, the negative regulator of *SIRT1*, resulting in functional changes that were consistent with increased *SIRT1* activity. These findings identify several potential targets for modifying adaptive immune responses.

Little is known about the direct link between human T cell-specific metabolic programs and function. Unlike other cell types, the link between T-cell function and metabolism is contradictory, and most studies have used murine T cells. Our study provides a detailed analysis of the energy requirements for human CD4[+] T cells at the metabolic, functional, and transcriptional levels. T-cell activation and proliferation are two metabolically and functionally distinguishable processes with distinct metabolic requirements. We confirm that mitochondrial ATP production is essential for T-cell activation and that their proliferation is associated with marked glucose uptake and glycolysis as the major source of ATP. Glucose deprivation inhibited T-cell proliferation but not activation, suggesting that efficient T-cell activation can occur in the absence of an external glucose source if their mitochondrial function is intact.

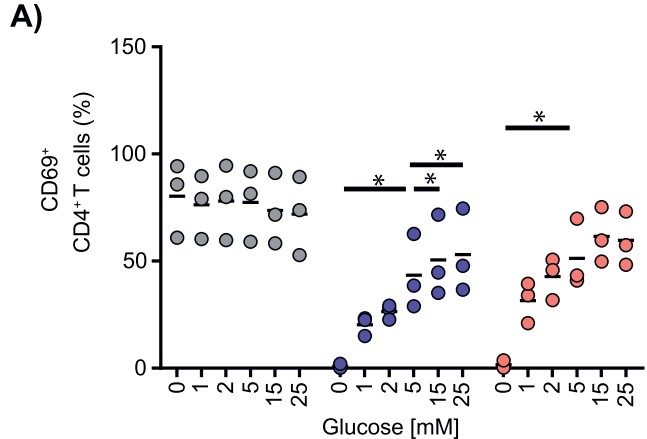

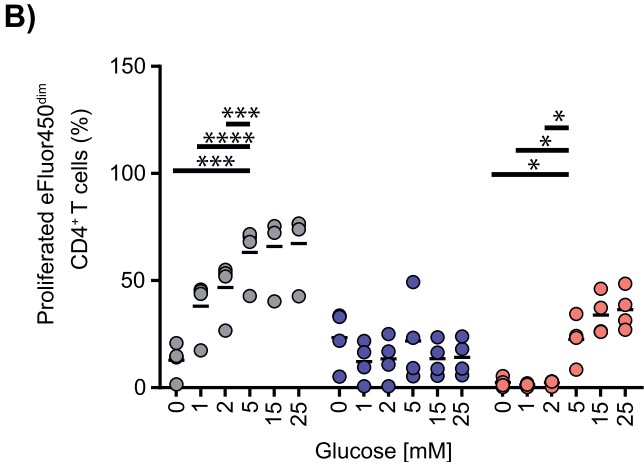

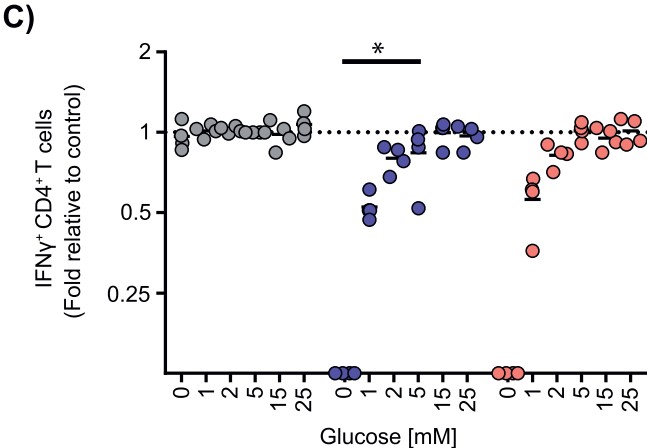

**Figure 5. Effects of glucose deprivation during mitochondrial stress on T-cell functions.**

**(A, B, C)** CD4⁺ T cells were cultured and stimulated for (A) 16 h to measure T-cell activation as the frequency of CD69⁺ cells, (B) for 72 h to measure T-cell proliferation, and (C) for 6 h to measure IFNγ production with the indicated glucose concentration without (black symbols) or with a high concentration (1 μM) for T-cell activation and cytokine production and low concentration (0.1 μM) for T-cell proliferation of the mitochondrial inhibitors oligomycin (blue symbols) or rotenone (red symbols). Each point represents one donor (n = 3).

Although human T cells appear to follow an established metabolic transition from OXPHOS to glycolysis for T-cell effector function, we provide several lines of evidence showing that T cells use ATP from various sources for their activation, proliferation and cytokine production. First, mitochondrial perturbation downstream of the activation phase significantly inhibited proliferation. Second, and importantly, combining the mitochondrial perturbation with glucose deprivation exacerbated the inhibitory effects of either perturbation on the activation and proliferation of T cells. In particular, rotenone combined with low glucose completely blocked T-cell proliferation with only moderate effects on T-cell activation. Stimulation of T cells at low glucose concentrations or in the absence of glucose decreased mitochondrial ATP production and T-cell proliferation, but did not affect T-cell activation. Thus, mitochondria are essential for T-cell activation and continue to provide energy for proliferation, whereas glucose is essential for proliferation and supports, but is not essential for activation. These data suggest an additional role of glucose metabolism in T-cell activation, and that the availability of glucose in the presence of mitochondrial stress equips the cell with increased metabolic flexibility and fitness. The independence of activation from glycolysis is also supported by previous findings (Sena et al, 2013).

A number of metabolic mediators, including $Ca^{2+}$ signaling and ROS production, are altered during T-cell activation (Sena et al, 2013; Weinberg et al, 2015). Although high levels of ROS lead to oxidative stress, ROS are also associated with the induction of signaling pathways (Mehta et al, 2017). We confirmed that ROS production was increased upon T-cell activation. However, unlike findings in murine T cells, an increase in ROS did not seem to be required for an efficient activation of human T cells. We found that resveratrol reduced ROS levels but did not affect T-cell activation. The inhibition of complex I and ATP synthase, and, thus electron transport chain function, increased mitochondrial ROS levels, but decreased T-cell activation. Our findings differ from those of a previous study using complex III knockout mice in which decreased ROS levels were linked to impaired T-cell activation (Sena et al, 2013). We suggest that the results of both studies may be reconciled by redox imbalance and electron transport chain dysfunction, resulting from the inhibition of complex I or complex III. The redox balance is important for sustained glycolytic flux and TCA engagement (Birsoy et al, 2015; Dimeloe et al, 2017). Our observation that FCCP partially restored rotenone-induced T-cell inhibition indicates that the redox balance is crucial for T-cell activation. It is also consistent with the engagement of the TCA cycle during T-cell activation.

Our findings also highlight a potential role of the IFN pathway, particularly OAS signaling, in T-cell activation. These pathways were impaired by short-term inhibition of mitochondrial ATP production in memory CD4⁺ T cells and either inhibition of mitochondrial ATP or complex I in naive CD4⁺ T cells. Oligomycin interferes with *OAS3* and *RNASEL* in innate immune cells (Wu et al, 2016; Yang et al, 2017;

---

Data are represented as mean value. Comparisons to the control conditions of 5 mM glucose were made using repeated-measures one-way ANOVA with a single pooled variance and uncorrected Fishers' least significant difference. *$P \leq 0.05$, **$P \leq 0.01$, and ****$P \leq 0.0001$.

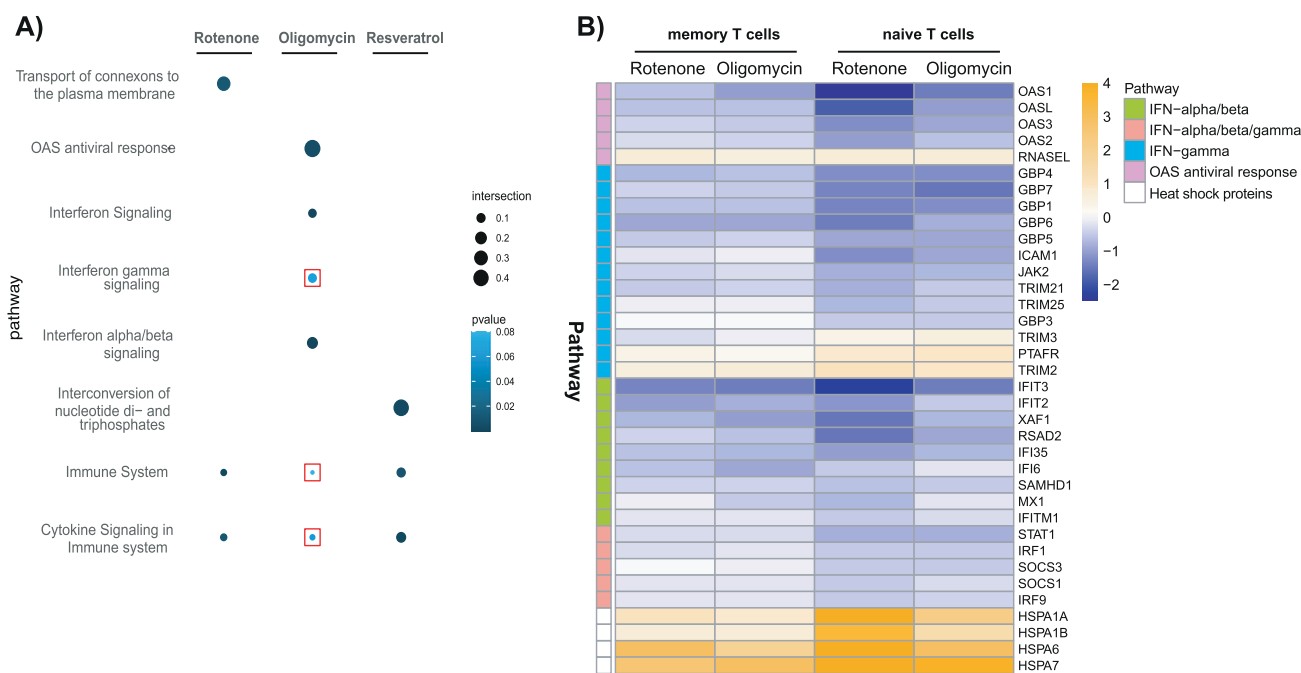

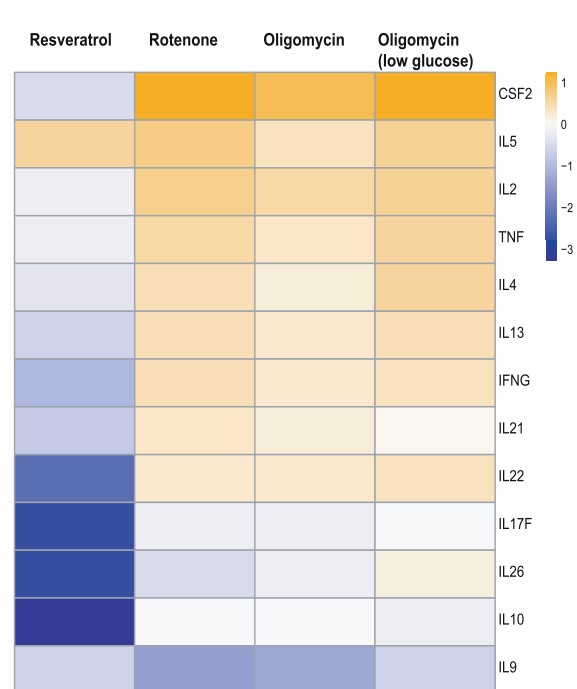

**Figure 6. RNAseq analysis of memory CD4⁺ T cells in the presence of metabolic inhibitors.**
**(A)** Dot plot showing the results of the pathway enrichment analysis of differentially expressed genes in the memory CD4⁺ T cells determined using gprofiler2. The size of the dot is proportional to the ratio of the differentially expressed genes in a particular Reactome pathway and the total number of genes assigned to that pathway (pathway coverage). The color of the dot shows the pathway enrichment significance (adjusted $P$-value). The pathways with adjusted $P$-values of >0.05 are indicated with a red box. **(B)** Heat map showing the $\log_2$ fold changes of the differentially expressed genes associated with the OAS antiviral response and IFN pathways as well as heat shock proteins in memory and naive T cells after exposure to rotenone and oligomycin. **(C)** Heat map showing the $\log_2$ fold change values of cytokine secretion genes in memory T cells with resveratrol, rotenone, and oligomycin, and oligomycin in the presence of 1 mM glucose.

Skrivergaard et al, 2019). Here, we show that oligomycin strongly down-regulated *OAS1-3* and *OASL* and up-regulated *RNASEL* in activated T cells. This is consistent with a block in the *OAS* gene response leading to stress-induced expression of *RNASEL*, which

requires OAS proteins and ATP for its activity. Active RNASEL amplifies the IFN response (Malathi et al, 2005) and is induced by a variety of stress conditions (Pandey et al, 2004). A number of genes that are up-regulated by type I IFNs, including several guanine binding protein

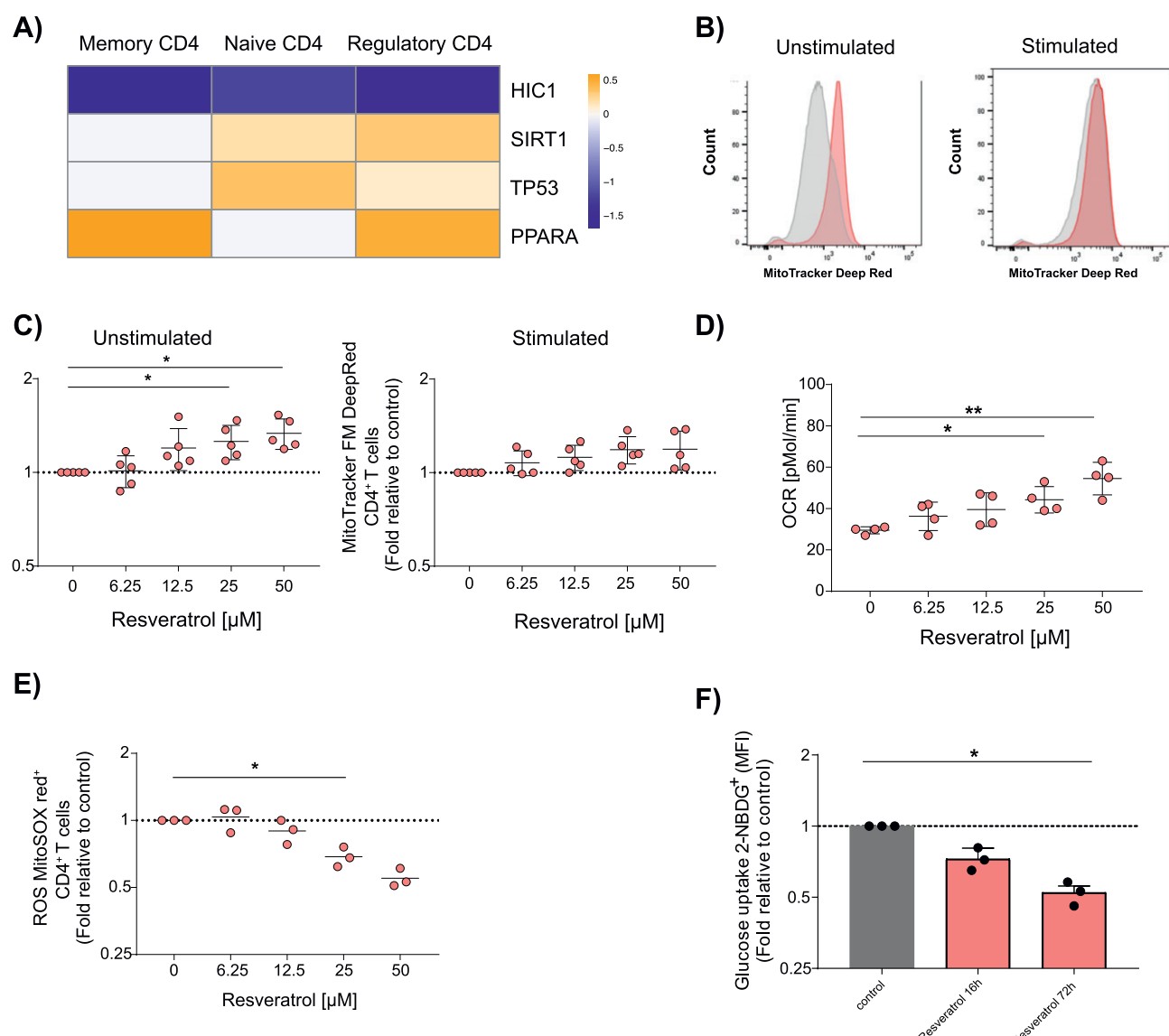

**Figure 7. Metabolic consequences of resveratrol on human CD4[+] T cells.**
**(A)** Heat map showing the $\log_2$ fold-change values of naive, memory and regulatory T cells when comparing the resveratrol treated samples to control conditions. HIC1 was transcriptionally down-regulated by resveratrol in all three CD4[+] T cell subsets. A concomitant increase in SIRT1 and downstream genes such as TP53 and PPARA in some of these T cells subsets was also observed. **(B)** Representative flow analysis histograms of MitoTracker Deep Red FM in CD4[+] T cells after 16 h in anti-CD3/CD28 unstimulated and bead-stimulated conditions in the presence (red) or absence (grey) of 50 $\mu$M resveratrol and 5 mM glucose. **(C)** Quantification of the mitochondrial mass changes shown in (B). The fold-values of the MitoTracker Deep Red FM Median fluorescence intensity for increasing resveratrol concentrations relative to no resveratrol for unstimulated (left) and stimulated (right) are shown as mean ± SD. **(D)** The spare respiratory capacity was measured using an extracellular flux analyzer 48 h after stimulation with anti-CD3/CD28 beads together with increasing resveratrol concentrations in the last 16 h of incubation. **(E)** Mitochondrial reactive oxygen species production was measured using MitoSOX Red at different resveratrol concentrations. **(F)** Cellular glucose uptake was measured in the presence of 50 $\mu$M resveratrol after 16 and 72 h using 2-NBDG. Comparisons to the control conditions without inhibitor were made using repeated-measures one-way ANOVA, with individual variances computed for each comparison. *$P \leq 0.05$ and **$P \leq 0.01$.

genes, were down-regulated in memory and naive CD4[+] T cells stimulated in the presence of oligomycin or rotenone. Our in vitro stimulation was devoid of exogenous type 1 IFN, suggesting that inhibition of intrinsic T-cell IFN signaling is coupled to reduced RNASEL activity.

Murine cytokine-secreting effector T cells are supported by increased glycolytic flux at the epigenetic and posttranscriptional level (Cham et al, 2008; Blagih et al, 2015). Cytokines are secreted immediately after TCR ligation, but also hours or days post

activation. We revealed robust cytokine secretion in the presence of impaired mitochondrial respiration or glucose deprivation. Only resveratrol and the combination of blocking both mitochondrial respiration and glucose source impaired the production of all measured signature cytokines. These data highlight a general robustness of cytokine production in broad ranges of cellular stresses, as compared with T-cell activation and proliferation. Nevertheless, also cytokine production was strongly impaired when

overall sources of energy were restricted. The gene expression profile of memory T cells confirmed the effects of resveratrol on the cytokine signaling pathway, and we also observed a strong effect of resveratrol on the interconversion of nucleotide diphosphates and triphosphates. Enzymes in this pathway are required for the synthesis of nucleotides, which are essential for a variety of cellular metabolic processes, as well as RNA and DNA synthesis. It is unlikely that this pathway is involved in cytokine production, and it is more likely to provide a mechanism for the strong inhibition by resveratrol on T-cell proliferation.

Many beneficial health effects are attributed to resveratrol. These include anti-inflammatory properties, protection against oxidative cell damage, induction of tumor cell death, and reversing obesity-induced metabolic disturbance (de Ligt et al, 2015; Szkudelski & Szkudelska, 2015; Park et al, 2016). The proposed mechanisms of action of resveratrol include direct and indirect interference of mitochondrial pathways. The most frequently discussed targets of resveratrol action are complex I or ATP synthase (Gledhill et al, 2007; Gueguen et al, 2015), interference with mTOR, Sirt1-mediated AMPK activation, or inhibition of nuclear factor-κB signaling (Price et al, 2012; Ma et al, 2015; Park et al, 2016). It was also reported that resveratrol has opposite effects on T cells at high and low concentrations (Craveiro et al, 2017), a finding that we could not confirm. The previous study reported that 20 μM resveratrol increased SIRT1 expression and induced genotoxic stress with engagement of DNA damage response pathways. Concomitant activation of p53 in T cells was linked to metabolic reprogramming characterized by decreased glycolytic rates and a shift towards OXPHOS. The study also reported that resveratrol exerted antioxidant effects at high concentration, but increased ROS at low concentrations alongside an increased effector function measured by IFNγ production (Craveiro et al, 2017). Our data contradict these findings. Resveratrol did not affect T-cell respiration during extracellular flux analysis. We found that 25 μM resveratrol decreased ROS production, and we did not identify a decreased activation capacity at any resveratrol concentration. We also observed a dose-dependent decrease in cytokine secretion. A novel finding of our study was that resveratrol down-regulated HIC1, the negative regulator of SIRT1, in memory, naive, and regulatory CD4+ T cells. This is particularly interesting given that increased SIRT1 activity is a postulated effect of resveratrol and many of the effects of resveratrol observed in our study and in prior reports are consistent with increased SIRT1 activity.

Our findings are based on an in vitro stimulation model. We purposefully chose a model devoid of interference from innate immune cells so that we could elucidate the effects of metabolic perturbation that are intrinsic to CD4+ T cells. However, it is likely that the observed effects may vary when other stimuli are used to activate T cells and in an in vivo setting. Moreover, although we identified a number of pathways and genes that were affected by the metabolic perturbations, we could not prove their direct involvement in the T-cell processes by specific knockdown or overexpression.

T cells must have a certain adaptive capacity if they are to be effective in sites of inflammation and infection. We found that T-cell activation and cytokine secretion were particularly robust in the presence of low glucose, suggesting a larger tolerable range of glucose fluctuations for immune function than was previously reported for mice. Only T-cell proliferation decreased in a dose-

dependent manner. Although the T-cell activation and effector functions were tolerant to glucose fluctuations, they were strongly compromised when glucose availability was reduced in the presence of mitochondrial stress. This is likely to be due to a lack of fuel to supply mitochondrial processes at a time when these are already compromised. Hence, conditions associated with mitochondrial stress together with a reduced ability to uptake glucose, such as in diabetes or in a tumor environment, may be prone to inefficient immune responses. Therefore, we suggest that increasing glucose availability to immune cells along with improving mitochondrial function may be useful therapeutic adjuvants to improve immune activity.

# Materials and Methods

### Study population, and isolation and culture of human CD4+ T cells

Healthy adult donors were recruited from the Munich Diabetes Bioresource study (ethical approval number 5049/11; ethical approval board: Technical University of Munich, Fakultät für Medizin). Peripheral blood mononuclear cells were isolated from heparinized blood by density gradient centrifugation over Ficoll-Paque (GE Healthcare). Human CD4+ T-cell enrichment and, when required, CD25+ cell depletion was performed by magnetic-activated cell sorting. CD4+ MicroBeads and human CD25+ MicroBeads II (Miltenyi Biotec) were used according to the manufacturer's protocol. Isolated T cells were cultured in glucose-free RPMI 1640, supplemented with 5% human male AB serum, 2 mM L-glutamine, 1% penicillin–streptomycin solution, and glucose at concentrations ranging from 0 to 25 mM. Cells ($10^6$/ml) were cultured in 96-well plates (Costar) and stimulated using anti-CD3/CD28 Dynabeads (Thermo Fisher Scientific) at a 1:4 bead-to-cell ratio.

### Bioenergetic measurements

The Seahorse XF Assay was performed using an extracellular flux analyzer (Seahorse XF96, Agilent). CD4+ T cells ($2 \times 10^5$/well) were cultured and stimulated with anti-CD3/CD28 Dynabeads (Thermo Fisher Scientific) in 96-well culture plates (Costar) for 48 h in RPMI 1640, and then were transferred to XF cell culture plates. Oxygen consumption was measured and the bioenergetics partitioned by injecting 2 μM oligomycin to inhibit ATP synthase, 1.5 μM FCCP to uncouple and drive maximal substrate oxidation, 2.5 μM rotenone to inhibit complex I, and 2.5 μM antimycin A to inhibit complex III in real time. Glycolysis was monitored with ECAR values and distinguished from non-glycolytic acidification by addition of the glycolytic inhibitor 2-deoxyglucose (2-DG) at 100 mM. The OCR and proton production rate are reported as pmol/min.

### Cell stimulation and flow cytometry

Cells were stimulated with anti-CD3/CD28 Dynabeads for 6 h to measure cytokine production, 16 h to measure activation markers, or 72 h to measure proliferation. To assess cell viability, the T cells were stained using a Zombie NIR fixable viability kit (BioLegend) or

7-aminoactinomycin D viability staining solution (BioLegend) according to the manufacturer's protocols. For cytokine production, brefeldin A (10 μg/ml; Sigma-Aldrich) was added for the last 4 h of culture. The cells were fixed and permeabilized using Cytofix/Cytoperm solution (BD Bioscience) according to the manufacturer's protocol. The cells were subsequently stained with anti-CD4-FITC (BioLegend) and intracellular staining was performed with anti-IL-4-Brilliant Violet 605 (BioLegend), anti–IL-17A-Brilliant Violet 510 (BioLegend), anti–IL-22-eFluor450 (eBioscience), anti–IFNγ-PE-Cy7 (BD Bioscience), and anti–IL-21-PE (eBioscience). For activation, cells were stained with anti–CD4-PerCP (BD Pharmingen), anti–CD69-FITC (BD Pharmingen), anti–CD25-PE (BD Pharmingen), and anti–CD154-Brilliant Violet 605 (BioLegend), and fixed with 1.5% formalin in $PBS^{-/-}$. For proliferation, cells were labeled with the dye eFluor450 (10 $\mu$mol/l for $10 \times 10^6$ cells/ml) (eBioscience) before stimulation. After 72 h, cells were stained for CD4 (anti–CD4-PerCP; BD Pharmingen), and processed for flow cytometry.

Glucose uptake in anti-CD3/CD28 Dynabead–stimulated $CD4^+$ T cells was measured using the labeled glucose 2-NBDG (Thermo Fisher Scientific). The cells were cultured in 5 mM glucose and starved in glucose-free RPMI 1640 for the last 2 h before measurement. Then, 50 $\mu$M of 2-NBDG was added and the cells were incubated for 10 min at 37°C. MitoSOX Red Mitochondrial Superoxide indicator (Thermo Fisher Scientific) was used to detect cellular ROS production in live cells. For this, 5 $\mu$M MitoSOX Red was added to the assay medium before the cells were processed for flow cytometry (Sena et al, 2013). MitoTracker Deep Red FM (Invitrogen) was used to measure the cellular mitochondrial content. $CD4^+$ T cells were stimulated with anti-CD3/CD28 Dynabeads for 24 h. Cells were stained with MitoTracker Deep Red FM at a final concentration of 5 nM for 15 min at 4°C together with anti–CD4-Brilliant Violet 510 (BD Pharmingen), anti–CD25-Brilliant Violet 421 (BioLegend), anti–CD45RA-FITC (BioLegend), and anti–CD127-PE-Cy7 (BioLegend).

For all flow analyses, the stained cells were acquired immediately using a flow cytometer (LSR Fortessa II; Becton Dickinson) and were analyzed with FlowJo software (Version 10; Treestar Inc.).

### RNA isolation and sequencing

RNA was isolated using the miRNeasy micro Kit (QIAGEN) according to the manufacturer's protocol. mRNA was isolated by poly-dT pull down with strand-specific RNA-Seq library preparation. Next-generation sequencing was performed using the Illumina platform with 30 million reads per sample (single end 75 bp). Fastq files for each sample were aligned to the human transcriptome (Ensembl ver 91) using STAR (version 2.5.4a). Reads that mapped to more than one location in the transcriptome were discarded. The resulting bam files were used to generate a count matrix using featureCounts.

### Data and statistical analyses

Comparisons between groups and conditions were performed using GraphPad Prism 7 software (GraphPad Software). Comparisons were conducted using repeated-measures one-way analysis of variance (ANOVA) with individual variances computed for each comparison. One-way ANOVA or paired $t$ tests for paired samples were also used as appropriate. Two-tailed $P$-values of < 0.05 were considered significant. Differentially expressed genes were identified via the R package DESeq2. The resulting $P$-value was then corrected for multiple testing using the Benjamini–Hochberg procedure. Genes that were significant after correction ($P \le 0.05$) and had a fold-change outside the range of $2^{-0.5}$ to $2^{0.5}$ were considered differentially expressed. The R-package EnhancedVolcano was used to visualize the results of differential gene expression analysis (Blighe et al, 2021). The gprofiler2 package was used to ascertain if the differentially expressed genes were enriched for a particular gene ontology term/KEGG pathway/mammalian phenotype term. g:Profiler (Raudvere et al, 2019; Kolberg et al, 2020) was used to assess the functional enrichment of differentially expressed genes.

## Data Availability

Processed RNAseq data in the form of counts matrix are deposited at the European Genome-Phenome Archive (study accession no. EGAS00001005565 at https://ega-archive.org/datasets).

## Supplementary Information

## Acknowledgements

We thank all physicians and study nurses at the Institute of Diabetes Research, Helmholtz Zentrum München, Munich, Germany, who helped in the recruitment of participants in the Bioresource study, particularly Melanie Bunk and Yvonne Kriesen. This work was supported by grants from the Juvenile Diabetes Research Foundation (JDRF-Nos 17-2012-16, and 17-2012-593) and by grants from the German Federal Ministry of Education and Research (BMBF) to the German Center for Diabetes Research (DZD e.V.).

### Author Contributions

L Holthaus: conceptualization, data curation, formal analysis, validation, investigation, visualization, methodology, project administration, and writing—original draft, review, and editing.
V Sharma: data curation, software, formal analysis, validation, investigation, visualization, and writing—original draft, review, and editing.
D Brandt: methodology.
A-G Ziegler: conceptualization, resources, supervision, funding acquisition, and writing—original draft.
M Jastroch: conceptualization, resources, data curation, formal analysis, supervision, and methodology.
E Bonifacio: conceptualization, resources, data curation, supervision, funding acquisition, validation, investigation, project administration, and writing—original draft, review, and editing.

## Conflict of Interest Statement

Ethical Standards: The Bioresource study was approved by the local institutional review board of the Technical University of Munich, Fakultät für Medizin, Ethikkommission (No. 5049/11) and was performed in accordance with ethical standards. Informed consent: Written informed consent was obtained prior to Bioresource study inclusion.

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
