## [Reviewer comments · Life Science Alliance]

Life Science Alliance

Functional and metabolic fitness of human CD4+ T lymphocytes during metabolic stress

Lisa Holthaus, Virag Sharma, Daniel Brandt, Anette-Gabriele Ziegler, Martin Jastroch, and Ezio Bonifacio

DOI: <https://doi.org/10.26508/lsa.202101013>

Corresponding author(s): Ezio Bonifacio, Technische Universität Dresden

Review Timeline:

Submission Date:	2021-01-07
Editorial Decision:	2021-03-16
Revision Received:	2021-06-24
Editorial Decision:	2021-07-12
Revision Received:	2021-09-11
Accepted:	2021-09-13

Transaction Report:

March 16, 2021

Re: Life Science Alliance manuscript #LSA-2021-01013-T

Prof. Ezio Bonifacio
TU Dresden
Center for Regenerative Therapies Dresden
Dresden
Germany

Dear Dr. Bonifacio,

Thank you for submitting your manuscript entitled "Functional and metabolic fitness of human CD4+ T lymphocytes during mitochondrial and nutrient stress" to Life Science Alliance (LSA). We apologize for this extended and unusual delay in getting back to you. The manuscript was assessed by expert reviewers, whose comments are appended to this letter. Based on the reviewers' comments, we would like to invite you to submit a revised version of the manuscript to LSA.

As you will note from the accompanying reviewers' comments, while the reviewers are enthusiastic about these findings, they also raise a number of concerns that we agree with. All of the reviewers concerns will have to be addressed prior to further consideration of this manuscript at LSA.

Thank you for this interesting contribution to Life Science Alliance. We are looking forward to receiving your revised manuscript.

Sincerely,

Shachi Bhatt, Ph.D.

Executive Editor

Life Science Alliance

<https://www.lsjournal.org/>

Interested in an editorial career? EMBO Solutions is hiring a Scientific Editor to join the international Life Science Alliance team. Find out more here -

https://www.embo.org/documents/jobs/Vacancy_Notice_Scientific_editor_LSA.pdf

B. MANUSCRIPT ORGANIZATION AND FORMATTING:

Reviewer #1 (Comments to the Authors (Required)):

This paper reports a detailed analysis of human T cells stimulated with a strong activator (anti-CD3/28 beads) in response to low glucose with and without, ATP synthase inhibitor, complex 1 ECT inhibitor and resveratrol. While some results are confirmatory, other results are novel, and present

the advantage of a side by side comparison.

Interpretation of results obtained with resveratrol is problematic due to its pleiotropic effect. Even within this paper, Sup. Fig 2A shows it as inhibiting mROS production, while the text p. 6 indicates that it inhibits the F1 subunit of ATP synthase.

More importantly, this paper is not well-written. Numerous inaccuracies and inconsistencies (topped by Fig. 7 that does not correspond to the text) make it difficult to evaluate the significance of the findings.

The concentration of glucose should be clearly indicated for each experiment

Fig. 2 shows results when rotenone, oligomycin and resveratrol were added only during the Seahorse assays. What would happen if they were added to the cultures from $t = 0$?

OCR measures oxygen consumption, it is not a measurement of mt ATP production as stated in the text.

Supl. Fig 3A: why is CD69 expression not higher in CD45RA+ T cells than in CD45 RA+? A FACS plot should show how CD69 expression is measured.

Sup Fig. 2B. does not show representative measurements. This figure (or text) should be used to explain what is shown in Fig. 2. ECAR is not proton production rate (extra-cellular acidification rate)

Fig. 4A. there is a discrepancy between the legend (presence / absence) and the graphs (4 increasing doses of each inhibitor). What are the comparisons made to? Where are the controls? p. 7 T cell proliferation is also shown in Sup. Fig. 4B and IFN γ in Sup. Fig. 4C. FACS plot should be shown for IL-4 and IL-21, which are more difficult to detect.

p. 8. In contrast, resveratrol [50 μ M] significantly and consistently decreased the production of IFN γ , IL-21 ($p = 0.034$), and IL-22 (Figure 4), and these reductions were dose-dependent (Supplementary Figure 6C). This should be Fig. 4D.

p. 10 Sup. Fig. 10 should be 9.

Fig. 6. Why does rotenone does not show the IFN pathways in Fig. 6B when it shows similar down regulation of IFN-related genes in Fig. 6A?

Fig. 7. What is the difference between Fig. 7A naïve and memory and sup. Fig. 9C? Contrary to what is stated in the text p. 10, Fig. 7a does not show: "The negative regulator of SIRT1, HIC1 was transcriptionally downregulated by resveratrol in all three CD4+ T cell subsets (Figure 7A)".

The rest of Fig. 7 does not correspond to what is described in the text.

Do figures need titles that are different from the title in the legend?

Reviewer #2 (Comments to the Authors (Required)):

Holthaus et al define biochemical needs of CD4+ T cells undergoing activation. As they rightly point out, the majority of existing data is from mice models, and experiments conducted on human samples are certainly an added value. To achieve their objectives, author induce T-cell activation in the presence of 3 metabolic inhibitors or varying glucose concentration. Some of the data they generate confirm paradigms that have been already demonstrated several times, also in humans, and therefrom the main findings of the paper can not considered completely novel. However, some sections are original and improve current knowledge. Nonetheless, I have major concerns on the organization of the result section and the presentation of the results which are often difficult to follow.

In particular

MAJOR

- Authors tried many different conditions and measured several readouts. However, the narrative is sometimes poor, giving the impression that authors are just showing everything they found, as a sort of list. This renders very difficult to grasp the most important messages of the paper (they are highlighted in the discussion which is well written, but the results section should guide the reader), Authors should just focus on most important and novel results or re-organize the Results section.
- Few lines at the end of each Results paragraph summarizing the major findings could certainly help. Similarly, better explaining the rationale for passing from an experiment to another will improve the flow.
- Pag 6, author state to measure "ATP production"; this is not true, they measure Oxygen, which may not totally reflect ATP production.
- Oligomycin is inhibiting T-cell activation (Fig 4) at concentrations 100 lower than those required to inhibit OXPHOS (Fig 2). I would rather expect the inverse.
- The fact that the only condition having major effects on cytokine production is Resveratrol, which does not affect neither ECAR nor OCR measures, may suggest that it acts through mechanisms independent by its effects on ATP-synthase/metabolism?
- at the beginning of page 8 Authors start to mention CD4 T cell subsets and CD8 T cells. I think this is very confusing. Regarding CD8 T cells, I would cut this part or, if authors want to keep it, dedicate it, at the end of the paper, an ad hoc paragraph where they compare, for all measures performed, CD4 and CD8.
- it's unclear the rationale of the experiments described in the last part of the paragraph ending at page 8.
- at the beginning of page 9, authors described some effects already shown in Figure 4. These unnecessary repetition may confuse the reader.
- at the end of page 9, authors start describing some in-depth analysis on CD4 subsets (at a certain point they also include Treg...). It's peculiar the decision of performing half of the experiments on total CD4 T cells and half comparing different subsets. This as well is confusing. I suggest the authors to rather focus on 1 of the aspects (metabolic requirements of the CD4 compartment or comparison of CD4 subsets)
- At the end of page 10, author should not state "SIRT1 activity" as it is not directly measured

MINOR

- Figure 1A and 4B are showing similar things.
- Abstract: OAS should be spelled out
- Why 2C is described before 2B?
- As, in Figure 2, Oligomycin; Rotenone and Resveratrol are not affecting glycolysis, it's not clear why they affect glucose uptake (Figure 3). Please discuss

We thank the reviewers for their comments, which were very helpful and constructive. We have revised the manuscript and tidied up result presentation including figures according to the comments. A response to each comment is provided below.

Reviewer #1

This paper reports a detailed analysis of human T cells stimulated with a strong activator (anti-CD3/28 beads) in response to low glucose with and without, ATP synthase inhibitor, complex 1 ECT inhibitor and resveratrol. While some results are confirmatory, other results are novel, and present the advantage of a side by side comparison.

- Interpretation of results obtained with resveratrol is problematic due to its pleiotropic effect. Even within this paper, Sup. Fig 2A shows it as inhibiting mROS production, while the text p. 6 indicates that it inhibits the F1 subunit of ATP synthase.

We agree that unlike Oligomycin and Rotenone, the effects of Resveratrol are less well defined and pleiotropic. There are some reproducible effects in the literature such as anti-oxidant. To better point this out, Suppl. Figure 2A was modified slightly, and the text on page 6 was extended by defining resveratrol further as a potential activator of SIRT1.

- More importantly, this paper is not well-written. Numerous inaccuracies and inconsistencies (topped by Fig. 7 that does not correspond to the text) make it difficult to evaluate the significance of the findings.

Writing the manuscript was a challenge because of the complexity of the findings. We have substantially reworked the results section and hope that it is more coherent.

- The concentration of glucose should be clearly indicated for each experiment

The main manuscript was revised accordingly.

- Fig. 2 shows results when rotenone, oligomycin and resveratrol were added only during the Seahorse assays. What would happen if they were added to the cultures from $t = 0$?

Resting T cell metabolism is hardly measurable in the Seahorse assay. It becomes measurable if the cells are pushed toward proliferation and why it was added after 0 h. As shown in Figure 4, adding these at 0 h markedly inhibits activation and proliferation and we would expect to see very low OCR and ECAR. This was the case in early pilot experiments that were not further pursued and are, therefore, not shown.

OCR measures oxygen consumption, it is not a measurement of mt ATP production as stated in the text.

This is now more accurately stated.

- Supl. Fig 3A: why is CD69 expression not higher in CD45RA⁺ T cells than in CD45 RA⁺? A FACS plot should show how CD69 expression is measured.

We expect that the reviewer refers to the comparison of CD45RA⁺ vs. CD45RO⁺ T cells. These are heavily stimulated cells and we expect that CD69 is hitting maximum. Of interest, the CD45RA⁺ cells appear more affected than the CD45RO⁺ cells. Below is an exemplary FACS plot. However, Reviewer 2 has suggested that these data are disruptive. We have, therefore, removed them from the manuscript.

A) CD69⁺ Expression in CD45RO and CD45RA T cells in control conditions

B) CD69⁺ Expression in CD45RO and CD45RA T cells with 1 μ M Rotenone

- Sup Fig. 2B. does not show representative measurements. This figure (or text) should be used to explain what is shown in Fig. 2. ECAR is not proton production rate (extra-cellular acidification rate)

Suppl. Figure 2B now shows representative Seahorse measurements for the three inhibitors.

ECAR description is changed.

- Fig. 4A. there is a discrepancy between the legend (presence / absence) and the graphs (4 increasing doses of each inhibitor). What are the comparisons made to? Where are the controls? p. 7 T cell proliferation is also shown in Sup. Fig. 4B and IFN γ in Sup. Fig. 4C. FACS plot should be shown for IL-4 and IL-21, which are more difficult to detect.

We defined the absence of the inhibitor as the control condition. We now matched the description in the figure legend better with the description in the text.

The control condition is 0 μ M of inhibitor with 5 mM glucose concentration. The increasing doses of each inhibitor were investigated to understand the drug kinetics and to be able to relate it to T cell function and to evaluate the different functional sensitivities. The description in the figure legend (presence/ absence) is changed (see above).

Representative FACS plots for all cytokines are now shown (Supplementary Figure 6).

- p. 8. In contrast, resveratrol [50µM] significantly and consistently decreased the production of IFN γ , IL-21 (p = 0.034), and IL-22 (Figure 4), and these reductions were dose-dependent (Supplementary Figure 6C). This should be Fig. 4D. p. 10 Sup. Fig. 10 should be 9.

Figures should now be correctly referred to in the text.

- Fig. 6. Why does rotenone does not show the IFN pathways in Fig. 6B when it shows similar down regulation of IFN-related genes in Fig. 6A?

This is a quirk of pathway analyses While the heatmap does show downregulation of IFN-related genes across both the treatments – Oligomycin and Rotenone, not all of the genes shown in the heatmap met our threshold set for the adjusted p-value. These genes were not taken up for further enrichment analysis and this explains why IFN-related pathways do not show up as significantly enriched on treatment with Rotenone. This figure is now slightly revised as A and B were not in the correct order.

- Fig. 7. What is the difference between Fig. 7A naïve and memory and sup. Fig. 9C? Contrary to what is stated in the text p. 10, Fig. 7a does not show: "The negative regulator of SIRT1, HIC1 was transcriptionally downregulated by resveratrol in all three CD4+ T cell subsets (Figure 7A)". The rest of Fig. 7 does not correspond to what is described in the text. Do figures need titles that are different from the title in the legend?

We have modified Figure 7A. The text has been reworded.

Reviewer #2

-Authors tried many different conditions and measured several readouts. However, the narrative is sometimes poor, giving the impression that authors are just showing everything they found, as a sort of list. This renders very difficult to grasp the most important messages of the paper (they are highlighted in the discussion which is well written, but the results section should guide the reader), Authors should just focus on most important and novel results or re-organize the Results section.

We have now tried to better organize the results rection and provide a better reasoning in the logic of the experiments.

-Few lines at the end of each Results paragraph summarizing the major findings could certainly help. Similarly, better explaining the rational for passing from an experiment to another will improve the flow.

We hope we have been able to improve the flow.

-Pag 6, author state to measure "ATP production"; this is not true, they measure Oxygen, which may not totally reflect ATP production.

This is consistent with Reviewer 1 and adjusted within the text.

-Oligomycin is inhibiting T-cell activation (Fig 4) at concentrations 100 lower than those required to inhibit OXPHOS (Fig 2). I would rather expect the inverse.

We were also a little surprised at the relative sensitivities to Oligomycin. However, it was previously shown that T cell activation further requires important mitochondrial signaling molecules such as Ca²⁺ or mtROS. Our results further suggest the importance of a maintained redox balance for T cell activation as shown in suppl. Figure 7. This has been extended in the results section.

-The fact that the only condition having major effects on cytokine production is Resveratrol, which does not affect neither ECAR nor OCR measures, may suggest that it acts through mechanisms independent by its effects on ATP-synthase/metabolism?

Yes, this seems to be the case and one of the reasons we examined resveratrol further. We suggested a mechanism of action downstream of mitochondrial pathways. Hypermethylated in cancer 1 protein (HIC1) was downregulated in naïve, memory, and regulatory T cells (new Figure 7A). HIC1 is a known negative regulator of SIRT1. Therefore, we propose an indirect activation of SIRT1 by resveratrol. The positive effects observed by resveratrol mimic the activation of SIRT1. This is demonstrated by improved mitochondrial fitness. Mitochondrial biogenesis and spare respiratory capacity are increased, and mitochondrial ROS production is decreased. Glucose uptake in the presence of resveratrol is diminished and mimics caloric restriction on a cellular level.

These experiments highlight the T cell flexibility of using different ATP sources for cytokine production. Alone, impaired glycolysis and OXPHOS rates did not affect cytokine production. Instead, in combination, cytokine production is impaired. This suggests that cytokine production can retrieve and adapt ATP source to the current status of OXPHOS and glycolytic rate.

-at the beginning of page 8 Authors start to mention CD4 T cell subsets and CD8 T cells. I think this is very confusing. Regarding CD8 T cells, I would cut this part or, if authors want to keep it, dedicate it, at the end of the paper, an ad hoc paragraph where they compare, for all measures performed, CD4 and CD8.

We have now deleted the CD8⁺ T cell data and regulatory T cell data. We have now moved the memory and naïve comparison to just before the gene expression data.

-it's unclear the rationale of the experiments described in the last part of the paragraph ending at page 8.

We have now tried to give this rationale more clearly.

-at the beginning of page 9, authors described some effects already shown in Figure 4. These unnecessary repetition may confuse the reader.

We have now rewritten this to avoid repetition.

-at the end of page 9, authors start describing some in-depth analysis on CD4 subsets (at a certain point they also include Treg....). It's peculiar the decision of performing half of the experiments on total CD4 T cells and half comparing different subsets. This as well is confusing. I suggest the authors to rather focus on 1 of the aspects (metabolic requirements of the CD4 compartment or comparison of CD4 subsets)

We agree that this is awkward and incomplete. We have, therefore, removed this part.

-At the end of page 10, author should not state "SIRT1 activity" as it is not directly measured

We have rephrased the notion of SIRT activity.

MINOR

-Figure 1A and 4B are showing similar things.

Figure 1 A shows the proliferation capacity in CD4⁺ T cells with oligomycin added at 0 h and washed out after 16 h. This was meant to understand needs in the activation phase vs the transition to proliferation phase. Figure 4 technically shows similar results with the addition of the inhibitors throughout the whole culture period. However, it is dosed and in addition to Oligomycin, also Rotenone, Resveratrol, and glucose are shown.

-Abstract: OAS should be spelled out

Revised as suggested.

-Why 2C is described before 2B?

This has been revised and the order is now 2B before 2C.

-As, in Figure 2, Oligomycin; Rotenone and Resveratrol are not affecting glycolysis, it's not clear why they affect glucose uptake (Figure 3). Please discuss

Our data suggest that 48 h bead-stimulated T cells regularly run high on glycolysis. There is little effect in adding the inhibitors at that point. Figure 3 has the inhibitors present from the start of stimulation until measurement at 16 h. We expect that because the inhibitors lead to a decrease in proliferation, there is less glycolysis going on ie there is less glucose requirement and why there is less glucose uptake. The wording has been changed to make this a little clearer.

July 12, 2021

RE: Life Science Alliance Manuscript #LSA-2021-01013-TR

Prof. Ezio Bonifacio
TU Dresden
Center for Regenerative Therapies Dresden
Dresden
Germany

Dear Dr. Bonifacio,

Thank you for submitting your revised manuscript entitled "Functional and metabolic fitness of human CD4+ T lymphocytes during metabolic stress". We would be happy to publish your paper in Life Science Alliance pending final revisions necessary to meet our formatting guidelines.

- please upload your main and supplementary figures as single files
- please consult our manuscript preparation guidelines <https://www.life-science-alliance.org/manuscript-prep> and make sure your manuscript sections are in the correct order (COI, AC, and Acknowledgment section have to be part of the main manuscript text)
- please add your main, supplementary figure legends to the main manuscript text after the references section;
- we encourage you to revise the figure legends for figures S2 such that the figure panels are introduced in alphabetical order;
- Figure S2 appears blurry relative to the other figures
- please label the panel C in Figure S3
- please add a Running Title to our system
- please add ORCID ID for the corresponding author-you should have received instructions on how to do so
- please note that titles in the system and manuscript file must match
- please use the [10 author names, et al.] format in your references (i.e. limit the author names to the first 10)
- please add callouts for Figures S1A, B; S2A-G; S3A-C; S5A, B; S8A-C to your main manuscript text
- please remove highlights from page 3
- please add a separate Data Availability statement with accession information for the RNA-seq data
- on Page 6, you still refer to oxygen consumption rate (OCR) as a measure of mitochondrial ATP production, which was pointed out by both Refs and still needs to be corrected.

LSA now encourages authors to provide a 30-60 second video where the study is briefly explained. We will use these videos on social media to promote the published paper and the presenting author. Corresponding or first-authors are welcome to submit the video. Please submit only one video per manuscript. The video can be emailed to contact@life-science-alliance.org

A. FINAL FILES:

B. MANUSCRIPT ORGANIZATION AND FORMATTING:

****Reviews, decision letters, and point-by-point responses associated with peer-review at Life Science Alliance will be published online, alongside the manuscript. If you do want to opt out of having the reviewer reports and your point-by-point responses displayed, please let us know**

immediately.**

Sincerely,

Reviewer #1 (Comments to the Authors (Required)):

the paper has only been minimally revised and it fails to provide a coherent body of results that advances the field.

Reviewer #2 (Comments to the Authors (Required)):

The authors replied to the points I raised.

September 13, 2021

RE: Life Science Alliance Manuscript #LSA-2021-01013-TRR

Prof. Ezio Bonifacio
Technische Universität Dresden
Center for Regenerative Therapies Dresden
Dresden
Germany

Dear Dr. Bonifacio,

Thank you for submitting your Research Article entitled "Functional and metabolic fitness of human CD4+ T lymphocytes during metabolic stress". It is a pleasure to let you know that your manuscript is now accepted for publication in Life Science Alliance. Congratulations on this interesting work.

*****IMPORTANT:** If you will be unreachable at any time, please provide us with the email address of an alternate author. Failure to respond to routine queries may lead to unavoidable delays in publication.*******

DISTRIBUTION OF MATERIALS:

Again, congratulations on a very nice paper. I hope you found the review process to be constructive and are pleased with how the manuscript was handled editorially. We look forward to future exciting submissions from your lab.

Sincerely,
